# The Role of Anticoagulation in Tumor Thrombus Associated with Renal Cell Carcinoma: A Literature Review

**DOI:** 10.3390/cancers15225382

**Published:** 2023-11-13

**Authors:** Chelsey M. Williams, Zin W. Myint

**Affiliations:** 1Department of Internal Medicine, University of Kentucky, Lexington, KY 40536, USA; chelsey.williams@uky.edu; 2Division of Medical Oncology, Department of Internal Medicine, University of Kentucky, Lexington, KY 40536, USA

**Keywords:** renal cell carcinoma, tumor thrombus, anticoagulation, immunotherapy, tyrosine kinase inhibitors, nephrectomy

## Abstract

**Simple Summary:**

Tumor thrombus occurs when tumor cells extend into a blood vessel. An estimated 10% of kidney cancer cases are complicated by tumor thrombus, often invading the renal vein with extension to the inferior vena cava. Up to 1% have tumor cells extending to the heart. The standard of care for these patients is surgical removal of the kidney tumor and the tumor thrombus. Research focuses on surgical techniques, imaging methods, and molecular markers for prognosis. The full benefit of anticoagulation remains controversial in these cases, considering unknown benefits and bleeding risk during tyrosine kinase inhibitor therapy. In this literature review, we summarize known data regarding the use of anticoagulation in the setting of kidney cancer and tumor thrombus.

**Abstract:**

Tumor thrombus (TT) is a complication of renal cell carcinoma (RCC) for which favorable medical management remains undefined. While radical nephrectomy has been shown to increase overall survival in RCC patients, surgical interventions such as cardiopulmonary bypass (CPB) and deep hypothermic circulatory arrest (DHCA) utilized to perform TT resection carry high mortality rates. While it has been documented that RCC with TT is associated with venous thromboembolism (VTE) development, anticoagulation use in these patients remains controversial in clinical practice. Whether anticoagulation is associated with improved survival outcomes remains unclear. Furthermore, if anticoagulation is initiated, there is limited evidence for whether direct oral anticoagulants (DOACs), heparin, or warfarin serve as the most advantageous choice. While the combination of immunotherapy and tyrosine kinase inhibitors (TKIs) has been shown to improve the outcomes of RCC, the clinical benefits of this combination are not well studied prospectively in cases with TT. In this literature review, we explore the challenges of treating RCC-associated TT with special attention to anticoagulation. We provide a comprehensive overview of current surgical and medical approaches and summarize recent studies investigating anticoagulation in RCC patients undergoing surgery, targeted therapy, and/or immunotherapy. Our goal is to provide clinicians with updated clinical insight into anticoagulation for RCC-associated TT patients.

## 1. Introduction

Renal cell carcinoma (RCC) comprises 3% of all cancers [1]. It is projected that in 2023, 81,800 new cases of kidney cancer will be diagnosed, with 14,890 resulting in fatalities [2]. Up to 4–10% of RCCs are associated with tumor thrombus (TT) [3]. Among RCC cases with TT, renal vein invasion is most common, occurring in 10–18%, while involvement of the inferior vena cava (IVC) is seen in 4–23% [1,4]. An additional 1% of TT cases may extend into the right atrium [1,4,5,6,7]. The majority of renal cancers associated with TT are of clear cell type [8].

Clinical findings of RCC with TT are often insidious and nonspecific, such as palpable abdominal mass [7]. Additional findings indicative of TT include visible venous collaterals on the anterior abdomen or the emergence of new pulmonary emboli with deep vein thrombosis [6,9]. Patients with untreated RCC-associated TT had a median survival time of 5 months prior to the immunotherapy era, underscoring the necessity of prompt surgical intervention and medical management of TT [9]. Radical nephrectomy with thrombectomy remains the standard treatment [3]. Cancer-specific survival is 5 years following complete resection of TT in all grades of RCC without metastatic disease [10]. In RCC patients with metastatic disease and TT, patients treated with cytoreductive nephrectomy and thrombectomy had a median overall survival of 1 year [11]. Multiple case reports now document that immunotherapy has decreased the size of TT. This has allowed patients who were initially not deemed to be surgical candidates to undergo surgery [12,13].

The Mayo Clinic has a defined grading system for TT extension, with a scale of grade 0–IV [10]. However, isolating the variables required for studying RCC with TT remains challenging, with an example of this being the vast variability in TT grade. The variability in TT location, directionality of TT growth in relation to venous return, invasion of TT into local vessel wall, and concurrent bland thrombus formation adjacent to TT further complicate treatment. It is crucial to note that TT behaves differently from bland thrombus, as TT is composed of malignant cells instead of fibrin products [1]. TT can also travel to the lungs during the intraoperative period, posing a risk of patient decompensation.

Current National Comprehensive Cancer Network guidelines do not address anticoagulation in patients with RCC-associated TT, leaving the decision to initiate anticoagulation to the physician’s discretion. The use of anticoagulation with receptor tyrosine kinase (RTK) and multi-kinase inhibitors poses a bleeding risk and remains a clinical concern [9]. As novel therapies for RCC with TT emerge, further investigation is needed to determine the safety and benefits of anticoagulation for patients with RCC-associated TT.

## 2. Does Tumor Thrombus Burden Impact Survival?

Tumor thrombi are classified on a graded scale of 0–IV. Type 0 is limited to the renal vein, type I involves infiltration of the IVC less than 2 cm above the renal vein, and type II extends to the IVC below the hepatic veins. Type III is defined as the progression of TT up to the hepatic veins but below the diaphragm, and type IV extends above the diaphragm and includes the right cardiac atrium [4,14]. The absence of TT has been shown to have a statistically significant impact on improved survival outcomes [10].

While it is well documented that a complete lack of TT is associated with an improved prognosis, discrepancies have arisen regarding whether the total TT burden affects prognosis. Previous studies have demonstrated no significant difference in overall survival among patients with grade 0–IV TT [3,10]. Subsequent studies, such as those by Oltean et al., have found no correlation between TT formation and increased organ or lymph node metastases [9]. However, patients with grade III and IV TT were more likely to experience tumor progression. Patients with grade III and IV TT also showed increased median operative times, blood loss, and hospital length of stay [3]. Higher-grade TT also requires implementation of surgical techniques such as cardiopulmonary bypass (CPB) and deep hypothermic circulatory arrest (DHCA). While TT burden does not appear to impact overall survival, increased burden does play a key role in other factors that negatively affect the patient.

Additional factors of TT composition are being studied to determine whether the components of the TT itself affect survival and prognosis. In a recent study conducted by Jia et al., the effect of TT consistency on RCC patient prognoses was assessed [15]. This study included 190 patients classified as having either friable or solid TT. The consistency of TT was not found to impact prognosis. In contrast, a study performed by Weiss et al. found that patients with RCC and friable TT had a significantly shorter median overall survival than those with solid TT [16]. TT consistency was found to be an independent significant predictor of overall survival in RCC patients without distant or nodal metastases. Another study performed by Rabinowitz et al. aimed to characterize clinicopathologic differences and survival outcomes among patients with IVC TT based on whether the TT exhibited clear cell or non-clear cell histology [8]. The study found no differences between groups when assessing cancer-specific survival, recurrence-free survival, and overall survival. Surgical outcomes were also similar between the groups of patients undergoing nephrectomy.

## 3. Tumor Thrombus Growth Directionality Results in Higher Clot Burden

A retrospective analysis was conducted by Liu et al. at Peking University Third Hospital, which involved patients with RCC and TT who were undergoing surgery [17]. The purpose of the study was to investigate whether the directionality of TT growth was associated with a poor prognosis. Patients were stratified into two groups based on the direction of TT growth, with one group growing against the direction of venous return and the other growing in the same direction as venous return. The growth of TT against the direction of venous return was identified as an independent risk factor for worse progression-free survival. Additionally, this study demonstrated that individuals with TT growing against the direction of venous return exhibited higher proportions of TT clot burden, adherence to the IVC vessel wall, and an increased volume of bland thrombus adjacent to TT [17].

It was postulated by the authors of this paper that a TT growing against the direction of venous return results in sluggish blood flow, causing platelets and red blood cells to accumulate at the distal end of the TT, thus prompting the formation of lengthy bland thrombus. The propensity of TT growing against venous return prompts consideration as to whether anticoagulation should be initiated in these patients. Further investigation into the directionality of TT growth is necessary to determine whether anticoagulation is warranted for patients with TT growing against the venous return. This study is the sole exploration of TT growth directionality that we could find. 

## 4. Advancements in Diagnostic Imaging of Tumor Thrombus

The complexity and location of a TT require thorough pre-surgical planning aided by imaging. If the TT is found to invade the IVC wall, resection of this area of the IVC is necessary. However, when the TT is large enough to fill the entire IVC lumen, it becomes especially challenging to determine whether the TT is invading the IVC wall or simply causing compression. On CT images, distinguishing between a TT and bland thrombus is difficult since they appear similar. A biopsy with pathologic analysis, which is often not feasible, would be necessary to differentiate whether the clot is a TT or thromboembolic [1].

Magnetic resonance imaging (MRI) outperforms computed tomography (CT) imaging in distinguishing between TT and bland thrombi [18]. A recent study sought to establish an MRI diagnostic model for TT within the IVC to guide preoperative decisions. A total of 165 cases of RCC with TT localized to the IVC were retrospectively analyzed. All had MRI imaging performed to include the full IVC. The researchers found that TTs with irregular edge morphology were more likely to be associated with IVC wall invasion. Interestingly, the length of the bland thrombus provided a strong prediction of wall invasion [18]. Moreover, the length of the bland thrombus predicted the extent of IVC resection required during surgery. However, this study did not address the directionality of TT growth, even though growth against the direction of venous return might be associated with larger TT, as previously mentioned.

A study conducted by Sarswat et al. closely examined the role of FDG PET/CT in the diagnosis of benign versus malignant TT [19]. Among 135 cancer patients with RCC, hepatocellular cancer, and thyroid cancer, the SUV max was calculated from each TT. However, this study found that FDG PET/CT only exhibited a sensitivity of 62.97% and specificity of 77.77% for TT detection [19]. In a systematic review of 15 pre-operative imaging studies performed by Smith et al., it was concluded that a multi-technique approach—including CT, MRI, and reporting of the vascular radiographic parameter—is necessary to predict IVC wall invasion, thus assisting in surgical planning [20]. It appears that multimodal imaging is essential for surgical planning to effectively determine the appropriate approach for thrombectomy.

## 5. Surgical Approaches to Treatment of Renal Cell Carcinoma with Tumor Thrombus

Nephrectomy with thrombectomy is currently the standard in surgical care for patients with RCC and TT [3]. CPB with or without DHCA is recommended for patients with grade III–IV TT [4]. Schimmer et al. assessed the outcomes of patients undergoing DHCA [4]. Out of seven patients with RCC and TT, all patients underwent surgery with DHCA. The most common complication was excessive bleeding, attributed to the accessory venous collaterals formed secondary to chronic IVC obstruction from the TT location [4]. Balancing bleeding and thrombosis remains challenging in the operative setting. There is a risk for thrombus embolization during IVC manipulation, particularly in higher-grade TT. Inserting a Swan–Ganz catheter is recommended during CPB for monitoring [21]. Additionally, intraoperative trans-esophageal echocardiography (TEE) can be performed during resection of grade II–IV TT. In cases of acute intraoperative decompensation, prompt TEE is recommended if not already underway [21].

Studies concerning the surgical management and outcomes of RCC with TT are hindered by two key factors: small sample sizes and extended time spans ranging from several years to decades [3]. Because of this, independent prognostic factors and surgical outcomes of radical nephrectomy with thrombectomy remain ill-defined [3]. A study conducted in Beijing, China, involving 121 patients with RCC and TT, aimed to identify these independent prognostic factors across varying TT extensions [3]. TT levels were 0 in 25 patients, I in 20, II in 50, III in 12, and IV in 14. Overall survival was noted to be poor in all patients. A total of 51 patients (42%) died within the 24-month follow-up period and 69 patients (57%) experienced recurrence or metastases. Consistent with prior studies, TT level was not shown to affect overall survival; however, tumor necrosis and sarcomatoid differentiation had a direct, negative effect on overall survival [3].

Laparoscopic approaches may provide a better safety profile to treat RCC with TT. Fourteen patients who underwent laparoscopic radical nephrectomy in the setting of RCC were retrospectively identified, four of whom had TT localized to IVC. It was deemed that for these four patients, laparoscopic nephrectomy with thrombectomy was safe and effective [22]. A matched retrospective cohort study further explored these findings, this time with 324 patients [23]. Patients underwent either robotic, laparoscopic, or open radical nephrectomy with venous thrombectomy. Overall survival, tumor specific survival, local recurrence-free survival, and metastasis-free survival were assessed. The open radical nephrectomy group had the highest complication rate and longest postoperative hospital stay. However, there was no difference in overall survival, tumor-specific survival, and metastasis-free survival between groups. It is notable that patients who underwent minimally invasive surgery with laparoscopic or robotic techniques had better local recurrence-free survival compared with that of the open radical nephrectomy group [23]. A cross-sectional comparative study by Schernuk et al. demonstrated higher rates of major and minor postoperative complications with open nephrectomy for patients with RCC and TT [24]. Although there was a non-statistically significant higher postoperative mortality rate in the open procedure group, no differences in oncologic outcomes were observed between the groups [24]. Minimally invasive methods appear to improve local recurrence-free survival and reduce postoperative complications.

## 6. Molecular Biomarkers Impacting Prognosis

Investigation of molecular methods to determine prognosis of RCC with TT is currently underway. Shi et al. performed a genetic analysis of clear cell RCC with TT, aiming to identify distinct molecular features associated with the development of TT [25]. They established a six-gene classifier that effectively predicts patient survival. Furthermore, they identified that overexpression of the early growth response 2 (EGR2) transcription factor has an essential role in TT formation. In a separate study, Yang et al. identified a potential therapeutic target and biomarker, adherens junctions associated protein 1 (AJAP1), as being specific to RCC with TT [26]. They also observed increased immune cell infiltration and microsatellite scores that could serve as predictors of immunotherapy sensitivity. Another protein of interest is E-cadherin, which is well known to play a role in cell–cell adhesion. Additional studies now suggest that E-cadherin may contribute to platelet aggregation and clot stabilization [27]. While investigating friable versus solid TT as a predictor of clinical outcomes, Weiss et al. assessed the significance of E-cadherin [16]. They found a significantly greater amount of E-cadherin present in solid TT when compared with friable TT via immunohistochemistry. This suggests that increased levels of E-cadherin in TT tissue may contribute to improved overall survival. The authors suggest that many molecular factors contribute to TT consistency. Further investigation is warranted to confirm E-cadherin’s role in TT composition.

Additionally, a series of microRNAs has been identified in patients with clear cell RCC with the TT extending to the IVC: miR-221, miR-126, and miR-2126. Expression levels of these three microRNAs have been shown to be deregulated in thrombosis and embolic events [28]. This paper draws attention specifically to miR-221, noting that it has an association with recurrent thromboembolism. Interestingly, miR-221 has demonstrated its regulatory role in vascular endothelial growth factor 2 (VEGF-2) in both RCC and prostate cancer.

A case report by Labbate et al. performed multichannel immunofluorescence on TT tissue and demonstrated that TT was characterized by infiltration of immune cells, including CD + 8T cells, FoxP3+ regulatory T cells, and Batf3+ dendritic cells [13]. Interestingly, the TT had interspersed areas of strongly positive PD-L1 expression in stromal areas. In contrast, the primary RCC tumor tissue did not stain positive for PD-L1 expression or the other aforementioned markers noted in TT tissue. The primary tumor in this case was noted to be non-T-cell inflamed. When this patient received neoadjuvant immunotherapy with nivolumab and ipilimumab, there was complete pathologic regression of the TT within the IVC and left renal vein. The TT level was downgraded from grade IV to grade III. The primary tumor within the kidney remained stable in size. The differences of the microenvironment may explain the differing responses of the tumor cells to immunotherapy treatment [13].

Elucidation of biomarkers could pave the way for the development of molecular subtypes of RCC with TT. Ideally, these biomarkers could provide prognostic tools to assess the risk of progression and response to treatment. Biomarkers known to be directly involved in thromboembolism could aid in determining whether the patient would benefit from anticoagulation. Further exploration into TT biomarkers is warranted, as are studies of the local TT microenvironment and epigenetic regulation.

## 7. Impact of Anticoagulation Use with Tumor Thrombus

It remains debatable as to whether anticoagulation improves survival in cases of RCC with TT. A retrospective review of 153 patients with malignancy and TT treated with or without anticoagulation found no difference in survival between groups [28]. It is worth noting that this study was not exclusive to RCC; however, a substantial portion of the patients had either RCC or hepatocellular carcinoma. Patients already prescribed anticoagulation for other reasons were excluded from the study [29].

A recent study performed at Leiden University in the Netherlands evaluated the outcomes of anticoagulation in patients with RCC and TT [1]. The decision to administer anticoagulation, as well as the choice of anticoagulant, were determined at the treating physician’s discretion. The authors found that patients with RCC-associated TT frequently developed VTE, even while on anticoagulation, and experienced several major bleeding events [1]. While this might imply that the increased bleeding risk associated with anticoagulation outweighs the benefit of preventing VTE, drawing a definitive conclusion about anticoagulation’s utility from this study is challenging. Many of these patients were already on anticoagulation therapy for other comorbidities. Furthermore, some patients with TT were initiated on anticoagulation solely due to the presence of TT, and there was variability in which anticoagulation type each patient received. While certain individuals received low-molecular-weight heparin (LMWH), others were on vitamin K antagonists (warfarin) and none were on direct oral anticoagulants (DOACs). The study authors observed that patients who declined surgery were more likely to be started on anticoagulation compared with those who underwent surgery. Given the inconsistency of the multiple factors involved, including anticoagulation type and whether the patient underwent surgery, it remains uncertain whether the benefits of anticoagulation outweigh the risks. Further research is needed to determine which subgroups of patients with RCC and TT would benefit from anticoagulation therapy.

## 8. Tyrosine Kinase Inhibitors and Immunotherapy

It has been demonstrated that tyrosine kinase inhibitor (TKI) therapy improves overall survival in patients with RCC. TKIs are known to improve progression-free as well as overall survival. Interestingly, targeted therapies offer comparable benefits to patients over the age of 65 as they do to younger patients [9]. However, TKIs also come with an increased risk of thromboembolic events as well as bleeding [9]. Consequently, clinicians have maintained caution when considering initiating anticoagulation in patients with RCC-associated TT taking TKIs.

Cabozantinib, a drug that inhibits multiple tyrosine kinase receptors including VEGF, is approved for the treatment of RCC. Its efficacy was demonstrated in the CABOSUN and METEOR clinical trials. A multicenter study performed by Shayeb et al. examined the safety of cabozantinib and venous thromboembolism (VTE) formation. They found that DOACs may be safe for use with cabozantinib [30]. A total of 298 patients with RCC and TT receiving treatment with cabozantinib were divided into four groups: no anticoagulation, LMWH, DOACs, and warfarin. They found no difference in major bleeding events between the no anticoagulant, LMWH, and DOAC groups. The rate of recurrent VTE was similar across all anticoagulation groups. Notably, patients who did develop VTE had a median survival rate of 3.3 years, while those who did not have VTE had a mean survival of 6.0 years. Thus, VTE prevention was shown to have survival benefit in RCC-associated TT patients [30]. The risk of bleeding and thromboembolic events with immunotherapy, combined immunotherapy, and TKIs are not well studied. At present, the risk of these therapies with anticoagulation remains understudied.

Several case reports and limited studies have explored the effects of immunotherapy on RCC with TT (Table 1). In a prospective study of six patients with locally advanced RCC and IVC TT, both avelumab and axitinib were administered prior to nephrectomy and thrombectomy [31]. All six patients were found to have a decrease in size of both the primary tumor and the IVC TT. Notably, no adverse events (including bleeding) leading to surgery delay were reported [31].

An additional case of a 54-year-old female diagnosed with locally advanced clear cell RCC with level IV TT was initially determined to be unfit for surgery due to poor performance status [13]. Neoadjuvant immunotherapy with nivolumab and ipilimumab resulted in complete response of the IVC and renal vein TT. The size of the renal mass was unchanged. Due to the positive response of the TT to immunotherapy, she was able to undergo complete surgical resection and was disease-free more than one year after diagnosis.

In another case report, a 67-year-old male with grade IV TT and RCC was treated with dual checkpoint inhibitor immunotherapy with nivolumab and ipilimumab [32]. Following one year of treatment, his TT had decreased in size; however, it had not regressed from the right atrium. The patient underwent complete resection of the TT and renal mass. At present, he is one year post-surgery and continues to have no evidence of recurrence on imaging [32].

A 71-year-old female, also with grade IV TT and RCC, received combined immunotherapy with nivolumab and ipilimumab. She experienced TT progression into the right atrium while undergoing treatment [33]. Treatment was then switched to pazopanib monotherapy for five months. Following treatment with pazopanib, her renal mass decreased and the patient was able to undergo right nephrectomy [33]. Presurgical pazopanib for RCC with TT was further addressed in a retrospective review of seven patients. This study determined that all patients except for one experienced shrinkage of the primary tumor site as well as reductions in TT diameter and length [34].

A study of five patients performed by Yoshida et al. further evaluated the effect of neoadjuvant immunotherapy on RCC with TT [12]. Three patients received nivolumab plus ipilimumab and two received pembrolizumab plus axitinib. All patients had tumor shrinkage and two patients were found to have downstaging of IVC RCC TT. The average time of immunotherapy administration was 10 months [12]. Although these cases are limited in number, they highlight the potential benefits of immunotherapy administration prior to surgery.

A single, multi-center, phase II study (NAXIVA) investigated neoadjuvant axitinib administration for eight weeks in patients with both metastatic and non-metastatic clear cell RCC and TT prior to resection. Out of twenty evaluable patients, seven experienced a reduction in TT level. Among those who underwent surgery, seven out of seventeen had less invasive procedures than initially planned [36]. Additionally, a second phase II study found that neoadjuvant axitinib in patients with cT2a RCC resulted a significant decrease in tumor size in 12 out of 18 patients. The primary outcome measure was the number of patients who underwent partial nephrectomy, as opposed to total nephrectomy. The primary outcome was considered achieved in 12 patients, with many of these having their tumors downstaged. Effects on TT were not addressed in this study [37]. A third phase II clinical trial showed that 11 out of 24 patients with locally advanced nonmetastatic clear cell RCC who received presurgical axitinib experienced a reduction in primary renal tumor diameter. The other 13 patients had no progression of disease or tumor enlargement while on axitinib [38]. Table 1 provides details of additional neoadjuvant studies in RCC with TT.

It is notable that many of these patients treated with TKI and/or immunotherapy had subsequent downstaging of TT level. Efforts are currently underway to guide decision making for treatment options in this scenario, as demonstrated by Wu et al. The study was a retrospective analysis of tumor grade discrepancy between the primary tumor and TT in nonmetastatic clear cell RCC [39]. They found that upgrading and downgrading between primary tumor grade and TT grade is common. They created nomograms to facilitate patient selection for adjuvant therapy in patients with RCC and TT at high risk for disease recurrence. Tools such as this may be useful to physicians in deciding how to treat patients with RCC and TT [39].

Presurgical sunitinib has also been shown to reduce primary tumor size in patients with RCC, facilitating partial nephrectomy and preserved renal function in a clinical trial of 72 patients [40] (Table 2). Table 2 lists ongoing clinical trials for immunotherapy administration in RCC with TT. Although further prospective studies are necessary to fully assess the effects of neoadjuvant immunotherapy in patients with RCC and TT, the results of this study appear promising.

## 9. Choice of Anticoagulation: Warfarin, Heparin, or DOACs?

### 9.1. Warfarin

There are currently no ongoing randomized controlled trials comparing the safety and/or efficacy of anticoagulation in patients with RCC-associated TT. The limited size of study samples and the diverse nature of RCC cases with TT present challenges when attempting to establish broad guidelines for anticoagulation. It is clear that certain TKIs can interact with warfarin. Pazopanib, a multi-tyrosine kinase inhibitor approved to treat RCC, has been shown to prolong international normalized ratio (INR) and result in liver dysfunction, thus requiring close monitoring and even termination of both anticoagulation and pazopanib therapy [41]. This has also been shown with sorafenib [42]. Warfarin was shown to have an increased risk for bleeding with cabozantinib [30]. Given these interactions and risks, warfarin remains a suboptimal choice for anticoagulation in patients vulnerable to bleeding when undergoing TKI treatment. Moreover, its use demands strong patient adherence and close clinical monitoring.

### 9.2. Heparin

Multiple case reports documenting LMWH use in the setting of RCC with TT provide limited insight. For instance, a 70-year-old male with no past medical history was found to have anterior abdominal wall collateral vessels on physical examination and elevated inflammatory markers [9]. Right atrial thrombus was noted on echocardiogram along with renal mass as well as pulmonary and liver metastases. Long-term anticoagulation was initiated with dalteparin. Another case involving a 37-year-old female found to have RCC with IVC TT and extending to the right atrium, who was diagnosed during pregnancy, explored the challenges of treatment [6]. She was also found to have lupus anticoagulant antibodies. Indeterminant V/Q scan with known DVT prompted initiation of therapeutic enoxaparin. Nine days later, she was noted to have a right atrial mass on transthoracic echocardiogram and new extensive bilateral pulmonary emboli; she was then transitioned to a heparin drip. Six weeks after her diagnosis, she safely gave birth at 30 weeks pregnant and was then started on sunitinib.

A case of a 60-year-old male with a history of 20-pound weight loss, night sweats, fevers, and back pain with palpable abdominal mass was found to have RCC with TT into IVC and right atrium [7]. He had a large pulmonary embolism in the right main pulmonary artery. Imaging proved difficult to distinguish whether the emboli was an extension of TT or bland thrombus. It was determined that the bland thrombus was overlaying the TT; thus, the patient was started on enoxaparin. After undergoing extensive surgery, including open radical nephrectomy with CPB and DHCA, he was found to have brain metastases shortly after, leading to his death [7]. These case reports do not specify whether the patients experienced any bleeding complications from heparin initiation.

A review article exploring the management of IVC-localized TT suggests the potential need for anticoagulation and recommends LMWH as the preferred anticoagulant [21].

A phase I study examined the outcomes of dalteparin with sunitinib use in patients with metastatic clear cell renal carcinoma [43]. Notably, anti-factor Xa levels increased during combination treatment compared with dalteparin monotherapy, suggesting that sunitinib may increase the anticoagulation activity of dalteparin. Despite this, the therapeutic combination in patients was well-tolerated with few adverse events [43]. Therefore, LMWH should be initiated with caution, but may prove advantageous to warfarin if used with cabozantinib on the basis of the study performed by Shayeb et al.

### 9.3. Direct Oral Anticoagulants

The existing literature tends to focus on the safety of DOACs in patients with cancer-associated VTE rather than TT. Limited information is available regarding the use of DOACs with TT. The aforementioned cabozantinib study performed by Shayeb et al. suggests that DOACs might be as safe as LMWH when used concurrently with cabozantinib, as there were no significant differences in major bleeding events among the no anticoagulant, DOAC, and LMWH groups [30].

## 10. Conclusions

Treatment of RCC with TT remains understudied and warrants further prospective evaluation. Limited studies suggest that neoadjuvant treatment with immunotherapy can shrink TT, potentially offering an additional benefit in pre-surgical settings. Concurrent treatment with TKIs has also demonstrated a reduction in TT size. However, administering TKIs with anticoagulation poses a greater risk of bleeding. There is no well-documented overall survival benefit for patients with RCC-associated TT. Patients treated with anticoagulation can experience major bleeding events that can be life-threatening. Further research and subgroup analysis are necessary to determine which patients would clinically benefit the most from anticoagulation administered for TT treatment.

## Figures and Tables

**Table 1 cancers-15-05382-t001:** Neoadjuvant cancer-directed therapy for renal cell carcinoma with tumor thrombus.

Ref.	Patient Demographics	Location of Study	Diagnosis	Cancer-Directed Therapy	Tumor Thrombus (TT) Response	Primary Tumor Response	Surgical Intervention	Outcome
Case Reports								
Labbate et al. [13]	54-year-old female	Chicago, IL, USA University of Chicago	Locally advanced clear cell RCC with rhabdoid features and grade IV TT	Neoadjuvant nivolumab and ipilimumab	Complete pathologic response of TT localized to IVC and left renal vein Downstaging of level IV TT to level III There was viable residual TT within segmental renal veins of the renal sinus at time of resection	Renal mass remained stable Radiographic and immunopathologic signs of tumor resistance in primary kidney tumor	Left radical nephrectomy and IVC thrombectomy	Disease-free >1 year, no further systemic therapy administered
Master et al. [32]	67-year-old male	Atlanta, Georgia Emory University School of Medicine	Clear cell RCC with grade IV TT extending to right atrium	Neoadjuvant nivolumab and ipilimumab	Decrease in size of TT, but TT did not regress below the right atrium Histology demonstrated no residual viable tumor with the IVC TT	Size of primary tumor following administration of neoadjuvant immunotherapy not addressed	Radical nephrectomy and vacuum extraction of the thrombus from the IVC at the level of the renal vein	Disease-free >1 year from surgery and no evidence of recurrence
Nishimura et al. [33]	71-year-old female	Matsuyama, Japan Department of Urology Ehime University, Japan	Metastatic clear cell RCC with level IV TT extending to right atrium	Neoadjuvant nivolumab and ipilimumab for two cycles followed by 5 months pazopanib monotherapy	TT progressed from IVC into right atrium after 2 cycles immunotherapy, prompting switch to TKI therapy. Authors state this may have been pseudoprogression. No viable tumor cells in TT on pathologic analysis	Primary tumor decreased in size Formerly present lung nodules no longer detectable on imaging	Right nephrectomy and IVC thrombectomy	Disease-free at 1 year of time from surgery
Retrospective Studies								
Yoshida et al. [12]	5 patients, with mean age 65 years	Tokyo, Japan Tokyo Women’s Medical University Hospital	4 out of 5 patients with clear cell RCC, fifth patient type of RCC unknown TT level was IV in 1 of 5 patients and II in the remaining 4 of 5 patients	3 patients received nivolumab and ipilimumab 2 patients received pembrolizumab and axitinib	IVC TT level downstaged in 2 patients	All patients had a decrease in size of primary tumor.	Radical nephrectomy with thrombectomy performed in 3 patients	3 patients showed no signs of local recurrence or distant metastases 9 months after surgery 1 patient required radiation therapy for brain metastases
Zhang et al. [23]	16 patients, mean age 54 years	Guangzhou, China Sun Yat-sen University Cancer Center	14 of 16 patient with clear cell RCC	9 patients received neoadjuvant pazopanib + PD-1/CD-CIK cells immunotherapy 7 patients received neoadjuvant axitinib + PD-1/CD-CIK cells immunotherapy	TT volume reduced in all patients; 1 participant had downgrade of TT from IV to III; the volume of TT was decreased by an average of 72.82%	Overall tumor volume, including primary tumor, was decreased in all patients	10/16 patients underwent surgery; either radical nephrectomy, partial nephrectomy, or lymph node dissection	Long-term follow-up not addressed
Terawaka et al. [34]	7 patients with median age 67 years, 5 male, 2 female	Kobe, Japan Department of Urology, Kobe Graduate University, School of Medicine	All patients with level III or IV TT; 6 of 7 patients had metastases, 1 patient with sarcomatoid features, 1 patient with rhabdoid features, 2 patients with clear cell RCC, 1 patient with Xp translocation, and 2 patients of unknown type	All patients received presurgical pazopanib	TT decreased in both diameter and length in 6 of 7 patients; the patient with rhabdoid features did not respond to treatment.	Shrinkage of primary tumor observed in 6 of 7 patients. The patient with rhabdoid features did not respond to treatment.	3 patients underwent nephrectomy with IVC thrombectomy	The patient with rhabdoid tumor died of rapid disease progression After surgery, the 2 patients with clear cell RCC who underwent surgery maintained response to pazopanib
Okamura et al. [35]	16 patients with mean age of 69.8 years; 9 patients in treatment group	Kobe, Japan Kobe University Graduate School of Medicine	All patients with level III or IV TT 7 patients with clear cell RCC, 1 patient with sarcomatoid differentiation, and 1 unknown type	9 out of 16 patients received 12 weeks oral presurgical pazopanib and were compared with patients who only underwent surgery	All patients had shrinkage of TT; 7 patients gained lower TT level.	All patients who received presurgical pazopanib had shrinkage of primary tumor	7 patients underwent less-invasive surgeries	Long-term follow-up not addressed in this study
Prospective Studies								
Tobe et al. [31]	6 patients: 1 female, 5 male; mean age of 65 years	Kobe, Japan Department of Urology, Kobe University Hospital	5 out of 6 patients with clear cell RCC	All patients received combination avelumab and axitinib preoperatively	2 patients had downgrade of TT; 4 patients had stable TT throughout treatment	All patients had shrinkage of primary tumor	All underwent radical nephrectomy and IVC tumor thrombectomy	One patient had solitary lung metastasis 1 year after nephrectomy All patients alive 1 year after surgery

**Table 2 cancers-15-05382-t002:** Ongoing prospective clinical trials in renal cell cancer with tumor thrombus.

National Clinical Trial Identifier	Phase	Condition	Intervention	Status
05319015	II	Renal cell carcinoma with grade II–IV inferior vena cava tumor thrombus	Neoadjuvant pembrolizumab + lenvatinib	Active recruiting
05969496	II	Clear cell renal cell cancer, metastatic or non-metastatic, with inferior vena cava tumor thrombus	Neoadjuvant pembrolizumab + axitinib	Active recruiting

## Data Availability

No new data were created or analyzed in this study. Data sharing is not applicable to this article. The database used for the literature search was the National Center for Biotechnology Information (NCBI). Keywords employed for the literature search include renal cell carcinoma, immunotherapy, tyrosine kinase inhibitors, tumor thrombus, anticoagulation, heparin, DOAC, and warfarin.

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
