# Peer review of "The Role of Anticoagulation in Tumor Thrombus Associated with Renal Cell Carcinoma: A Literature Review"

_cancers, 2023, doi:10.3390/cancers15225382_

Round 1
Reviewer 1 Report
Comments and Suggestions for Authors
The article discusses the impact of anticoagulation and treatment methods for renal cell carcinoma (RCC) with tumor thrombus (TT) extending into the inferior vena cava (IVC). It mentions microRNAs, case studies, and the choice of anticoagulants like warfarin, heparin, and direct oral anticoagulants (DOACs). Neoadjuvant immunotherapy, tyrosine kinase inhibitors, and their interactions with anticoagulation are also covered. The article concludes that the management of RCC with TT requires further research, and patient selection for anticoagulation treatment remains a challenge.
Here are some improvements that can be made to the article:
- Document Structure: The article could benefit from a clearer and more logical structure. For example, you can divide the text into distinct sections for each topic discussed, such as "Introduction," "Treatment Methods," "Immunotherapy," "Anticoagulation," etc.
- Formatting: Currently, the text lacks paragraphs and proper spacing, making it less reader-friendly. It's important to break the text into clear paragraphs and separate different sections with appropriate headings.
- Clarity and Consistency: Ensure that sentences and paragraphs are clear and well-structured. Avoid using long and complex sentences that may confuse the reader.
- Grammar and Style Review: The text could be reviewed for correction of grammatical, punctuation, and style errors. For example, ensure that verbs and subjects agree and that language use is consistent throughout the article.
- Expansion and Depth: Some sections of the article could be further developed to provide more details or explanations. For example, you could provide specific examples or case studies to illustrate the concepts discussed.
- The discussion should start with a clear summary of the main results of the article so that the reader can quickly grasp the conclusions without having to go through all the details. The discussion should delve more deeply into the results presented in the article, identifying any significant trends or relationships between variables. It should also be clear whether the results support or contradict the article's initial hypotheses. The discussion should examine the clinical implications of the results and how they might impact medical practice or future research. For example, it could explore how the results may guide treatment decisions for patients with RCC and TT. Please include and discuss these interesting manuscripts: PMID: 37468393. PMID: 37109725.
- Consistency in Terminology and Abbreviations: Ensure that abbreviations and technical terms are used consistently. For example, if you use the abbreviation "TT" for "tumor thrombus," make sure to explain it clearly the first time it appears.
- Adding a Conclusion: The article could draw clear and concise conclusions from the data and discussions presented. This would help synthesize the key points of the article.
Reviewer 2 Report
Comments and Suggestions for Authors
The review titled "The Role of Anticoagulation in Tumor Thrombus Associated with Renal Cell Carcinoma: A Literature Review" provides a comprehensive examination of various aspects related to renal cell carcinoma (RCC) with tumor thrombus (TT). The review covers several key components, including the impact of TT on patient survival, the exploration of molecular biomarkers affecting prognosis, modern imaging diagnostic techniques, and contemporary treatment modalities.
The inclusion of a summary of previous reports on neoadjuvant therapies in RCC with TT in Table 1 is a valuable addition, as it consolidates case reports, retrospective studies, and prospective studies for quick reference. Similarly, Table 2, which outlines details about ongoing clinical trials on RCC with TT, provides a clear overview of the current research landscape in this area.
The review effectively concludes by pointing out the limited number of studies on TT in RCC and the necessity for further prospective evaluation, highlighting an important gap in the current body of research.
One suggested improvement would be to mention the databases used for the literature search and the keywords employed. Providing this information would enhance the transparency of the review process, helping readers understand the scope and rigor of the search strategy.
Overall, this is a well-written narrative review that explores an intriguing and understudied aspect of RCC, offering valuable insights and recommendations for future research.
Reviewer 3 Report
Comments and Suggestions for Authors
Authors should be congratulated for their work. The topic is interesting and intriguing. The role of anticoagulant treatment in RCC patients with vein thrombus represents an important topic, considering that these subgroup of patients need a thrombectomy+nephrectomy according to the current guidelines. The use of anticoagulant may be associated with an increased bleeding risk.
The manuscript is well structured, well-written and easly readable. Tables are of good quality. The references used are precise and focused. I suggest to include the within the introduction section the survival of metastatic patients treated with cytoreductive nephrectomy and thrombectomy, as shown by Sorce et al. (PMID: 36185586).
I recommed a grammar revision, as well as correction of all the typos throughout the manuscript.
Comments on the Quality of English Language
I recommed a grammar revision, as well as correction of all the typos throughout the manuscript.
Round 2
Reviewer 1 Report
Comments and Suggestions for Authors
The manuscript has been significantly improved compared to previous versions and is now an engaging and informative read. The topics discussed are extremely interesting, and the author has presented a unique and well-documented perspective.
Reviewer 3 Report
Comments and Suggestions for Authors
Authors should be congratulated for their work. The manuscript was improved according to the Reviewers' suggestions. The manuscript is suitable for publication in its current form.